# Adaptive Language-Guided Abstraction
# from Contrastive Explanations

**Andi Peng**
MIT

**Belinda Z. Li**
MIT

**Ilia Sucholutsky**
Princeton

**Nishanth Kumar**
MIT

**Julie A. Shah**
MIT

**Jacob Andreas**
MIT

**Andreea Bobu**
The AI Institute

**Abstract:** Many approaches to robot learning begin by inferring a reward function from a set of human demonstrations. To learn a good reward, it is necessary to determine *which features* of the environment are relevant before determining how these features should be used to compute reward. End-to-end methods for joint feature and reward learning (e.g., using deep networks or program synthesis techniques) often yield brittle reward functions that are sensitive to spurious state features. By contrast, humans can often generalizably learn from a small number of demonstrations by incorporating strong *priors* about what features of a demonstration are likely meaningful for a task of interest. How do we build robots that leverage this kind of background knowledge when learning from new demonstrations? This paper describes a method named ALGAE (Adaptive Language-Guided Abstraction from [Contrastive] Explanations) which alternates between using *language models* to iteratively identify human-meaningful features needed to explain demonstrated behavior, then standard inverse reinforcement learning techniques to assign weights to these features. Experiments across a variety of both simulated and real-world robot environments show that ALGAE learns generalizable reward functions defined on interpretable features using only small numbers of demonstrations. Importantly, ALGAE can recognize when features are missing, then extract and define those features *without* any human input – making it possible to quickly and efficiently acquire rich representations of user behavior.

**Keywords:** reward learning, language-guided abstraction, reward features

## 1 Introduction

When training robots to perform complex tasks – like watering plants in cluttered household environments (Fig. 1) – it is often useful to begin by specifying a reward function from which optimal robot behavior can be derived. Assigning rewards to trajectories typically requires extracting important state or trajectory-level features (e.g. *distance to goal* or *end effector orientation*), then using these features to compute scalar rewards. While it is sometimes possible for humans to directly specify reward functions in code, this process is challenging and prone to error, even for experts [1]. A slightly more general class of approaches uses manual specification of *feature functions* (either in code or with targeted supervision [2, 3]) followed by reward learning (e.g. with inverse reinforcement learning, or IRL [4]). But it can be challenging for users to identify and describe all features relevant to a task of interest (e.g. *the distance between the watering can and an optimal pouring height relative to the pot*), so reward learning with human-designed feature sets runs the risk of *feature under-specification* – situations in which important task-relevant information is unavailable to the reward function. An alternative family of *end-to-end* approaches, such as deep IRL [5, 6, 7] attempt to implicitly extract these features from user demonstrations, but can require

8th Conference on Robot Learning (CoRL 2024), Munich, Germany.

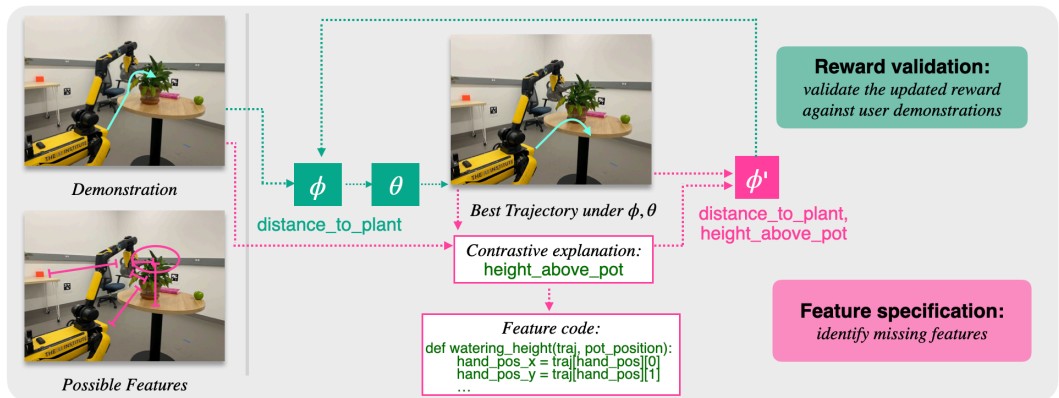

Figure 1: **Adaptive Language-Guided Abstraction from Contrastive Explanations** (ALGAE) alternates between two main stages: in *feature specification*, ALGAE expands the current feature set by identifying under-specified features of the current reward; then in *reward validation*, ALGAE learns an updated reward function defined on top of the new feature set and validates it can explain the user demonstrations. ALGAE results in more generalizable learned reward functions vs. baselines without manual feature specification, and can iteratively improve its own reward estimate given multiple under-specified features.

very large numbers of demonstrations to ensure that learned features are robust, generalizable, and insensitive to spurious features in the user demonstrations (e.g., *distance to apple*, etc. in Fig. 1).

Reliable, sample-efficient reward learning thus requires both identifying a discrete set of features that are salient to the task and learning how they parameterize a continuous reward function [8]. Fortunately, good features that explain human decision-making do not come from a blank slate. When learning (and planning), humans draw upon a wealth of prior experience to identify which features are meaningful and generalizable for acting in the world [9, 10]. Can we instill these priors into reward learning procedures without requiring human supervision at an impractical scale?

In this work, we propose to leverage the expressive human priors embedded in natural language text corpora. We describe an iterative framework for autonomously alternating between feature specification and reward learning. Our approach, called **A**daptive **L**anguage-**G**uided **A**bstraction from contrastive **E**xplanations, (**ALGAE**), leverages language models (LMs) in combination with user demonstrations to learn human-aligned reward functions informed by semantically-relevant features. ALGAE distills the reward learning problem into two components: 1) the *feature specification* problem of identifying missing reward features that are salient to the user's reward, and 2) the *reward validation* problem of finding and verifying a reward function that best explains demonstrated user behavior. By alternating between these two components, ALGAE iteratively builds increasingly rich representations of human decision-making while ensuring that learned rewards are explained by intuitive, semantically-meaningful features.

We empirically validate ALGAE's benefits across simulated and real-world robot tasks. ALGAE recovers missing reward features and produces trajectories that better align with users' ground truth reward compared to several baselines. Importantly, ALGAE can iteratively refine its own representation by sequentially identifying missing features and validating them against the demonstrations, thereby mitigating the *over-parameterization* problem exhibited by end-to-end approaches.

## 2 Related Work

Our work builds on several fundamental ideas in reinforcement learning and reward learning.

**Inverse reinforcement learning.** Inverse reinforcement learning (IRL) methods propose to learn the unknown objective function from observed actions in the environment, e.g., human demonstrations [4, 5, 11, 12]. Such methods suffer from identifiability issues [12, 13]. That is, given a constrained

number of demonstrations, multiple objective functions can explain the same observed behavior. This is due to expressive function approximators overfitting to a few demonstrations, a problem that is exacerbated in high-dimensional and messy state spaces such as in robot learning.

**Learning features from users.** A large body of work has studied how to elicit features (e.g. for use in RL) from human users. In this paradigm, learning good objectives is dependent on a set of carefully, hand-specified features that capture aspects of the environment or task that the user may care about [2, 14, 4, 1]. If selected well, this feature set introduces a well-formed inductive bias for facilitating more generalizable learning from few demonstrations. Such a feature set helps reduce the dimensionality of what is typically higher-dimensional states and trajectories, affording a better-shaped space for IRL to operate. But defining a good feature set is both challenging and laborious for experts and novices alike, and the chosen feature(s) may not be expressive enough to fully capture complex behavior in more ineffable robotic tasks [15]. Some studies explore having humans offer corrections to the agent using spatial labels [16], teleoperation with joysticks [17, 18], physical interaction [15], and natural language [19], but these approaches are difficult to scale as they involve expensive online human interventions.

**Learning features from language.** At a high level, most of the previous work can be described as designing methods for users to specify their priors about the environment and task to the robot. Recent work on language-guided abstraction ([20, 21]) instead aims to learn task-relevant features from language by leveraging the semantic priors embedded in language models [22] to best inform which environment elements are important to include in the state and which elements are not. These approaches are limited by the imitation learning framework—that is, there is no true way to learn an objective that *explains* the demonstrated behavior. As a result, these approaches reduce the number of human demonstrations required to learn preference-tailored generalizable policies, but may still require expensive, online interventions to correct for feature under-specification.

In this work, we extend the feature learning and language-guided frameworks to identify *under-specified* features using language-guided contrastive explanations between the demonstration and suboptimal trajectory, then iteratively learn reward functions using these features.

## 3  Problem Definition

We consider the problem of learning reward functions that capture the (unknown) preferences held by a human given a small number of user demonstrations and language.

**Preliminaries.** We model our problem as a Markov Decision Process (MDP) [23] $\langle \mathcal{S}, \mathcal{A}, \mathcal{T}, \mathcal{R} \rangle$, where $\mathcal{S} \in \mathbb{R}^d$ is the state space, $\mathcal{A}$ the action space, $\mathcal{T} : \mathcal{S} \times \mathcal{A} \times \mathcal{S} \to [0, 1]$ the transition probability distribution, and $\mathcal{R} : \mathcal{S} \times \mathcal{A} \to \mathbb{R}$ the reward function. A solution to the MDP is a policy $\pi : \mathcal{S} \to \mathcal{A}$ that specifies what actions the robot should take in different states. Following prior work [3], we assume that the robot operates under a parameterized linear reward function defined on the state, $\mathcal{R}_\theta(s) = \theta^\top \phi(s)$, where $\phi : \mathcal{S} \to \mathbb{R}^p$ is a feature vector and $\theta \in \mathbb{R}^p$ represents the reward weights on the features. The robot executes a task by following a state trajectory $\tau = \{s^0, ..., s^T\}$ given by the policy. To pick the best trajectory for the task, the robot optimizes the cumulative reward $\mathcal{R}_\theta(\tau) = \sum_{s^t \in \tau} \theta^\top \phi(s^t) = \theta^\top \phi(\tau)$, where $\phi(\tau) = \sum_{s^t \in \tau} \phi(s^t)$ is the total feature count along the trajectory, and solves:

$$\tau_{R_\theta} = \arg\max_\tau R_\theta(\tau) = \arg\max_\tau \theta^\top \phi(\tau) \ . \tag{1}$$

**Maximum Entropy Inverse Reinforcement Learning (MaxEnt IRL).** In practice, the reward function $\mathcal{R}_\theta$ is typically unknown to the robot or very challenging to manually specify. Thus, in IRL, the robot's goal is to *learn* $\mathcal{R}_\theta$ given human feedback like user demonstrations. Concretely, given human demonstrated trajectories $\mathcal{D} = \{\tau_i\}^{i=1...N}$, the robot interprets them as evidence for the human's preferred behavior and attempts to recover the reward parameters $\theta$ that explain the human's desired objective. We adopt the maximum-entropy framework for modeling human decision-making [12, 5], and model the human as a noisily-optimal agent that tends to choose demonstrations

in proportion to the exponentiated reward:

$$p(\tau \mid \theta) = \frac{e^{\mathcal{R}_\theta(\tau)}}{\int_{\bar{\tau}} e^{\mathcal{R}_\theta(\bar{\tau})} d\bar{\tau}} \quad \propto \exp(R_\theta(\tau)) = \exp(\theta^\top \phi(\tau)) \ . \tag{2}$$

Under this framework, the human is likely to act optimally and will generate suboptimal trajectories with a probability that decreases exponentially as the trajectories become lower in reward [12]. The unknown reward parameters $\theta$ can then be optimized via gradient descent on the objective:

$$\theta^* = \arg\max_\theta L(\theta) = \arg\max_\theta \sum_{\tau \in \mathcal{D}} \log p(\tau \mid \theta) \ . \tag{3}$$

To optimize this objective, we approximate the intractable integral in Eq. (2) using importance sampling, as in prior work [5, 3]. The robot then acts according to Eq. (1).

The features $\phi$ we choose to represent the reward $\mathcal{R}_\theta$ dramatically impact the reward functions that can be learned [2]. In the limited-data regime, this is true even when $\mathcal{R}_\theta$ belongs to an expressive function class (e.g., a neural network). Motivated by recovering a reward function $\mathcal{R}_\theta$ that incorporates human-like priors, we are interested in explicitly constructing *human-meaningful* features $\phi$. In this sense, we consider a feature set to be *under-specified* if it does not explicitly represent all the salient features relevant to the task, and consider a reward function to be under-specified if it uses an under-specified feature set as its basis.

# 4  Adaptive Language-Guided Abstraction from Contrastive Explanations

At a high level, ALGAE aims to identify the most likely set of preference weights $\theta^*$ and preference featurizations $\phi^*$ given the human demonstrations under $p(\phi, \theta \mid \tau_1, ..., \tau_N)$. The idealized joint optimization given demonstration $\tau$ thus takes the following form:

$$\theta^*, \phi^* = \arg\max_{\theta, \phi} p(\phi, \theta \mid \tau) \ . \tag{4}$$

We maximize this objective tractably by *iteratively* optimizing $\theta$ and $\phi$. Given an initialization of the feature vector $\phi^0$, ALGAE decomposes the learning problem into two subtasks: 1) performing *feature specification* by expanding each feature set $\phi^k$ into $\phi^{k+1}$ by querying for missing features in language, then 2) performing *reward validation* by finding an optimal reward weight $\theta^k$ for each $\phi^k$ and comparing its predictions to ground-truth trajectories.

## 4.1  Identifying missing features

In *feature specification*, we use an LM to identify missing features salient to the user's reward given a demonstration $\tau$ and the robot's current best estimate of the (under-specified) reward function $\mathcal{R}_{\theta^k}$. Once the missing feature has been identified, we query the LM for code to compute that missing feature and append the resulting feature function $\phi_i$ to the set of known features $\phi^k$ to obtain $\phi^{k+1}$.

Operationally, by fixing $\theta = \theta^k$, Eq. (4) becomes:

$$\phi^{k+1} = \arg\max_\phi p(\phi, \theta^k \mid \tau) = \arg\max_\phi p(\phi \mid \theta^k, \tau) \, p(\theta^k \mid \tau) = \arg\max_\phi p(\phi \mid \theta^k, \tau) \ , \tag{5}$$

where $p(\phi \mid \theta^k, \tau)$ may be estimated using an LM as described below.

**Querying for a missing feature.** The key insight of our approach is that language models have strong common-sense priors over salient task features (e.g., *ideal watering height above plant*) which we can leverage to approximate the probability $p(\phi \mid \theta^k, \tau)$ in Eq. (5). Concretely, we approximate the optimization Eq. (5) by first computing best estimated trajectory for the current reward $\tau_{R_{\theta^k}}$ using Eq. (1) and pair it with the user demonstrated trajectory $\tau$. We then prompt the LM with the contrastive pair of trajectories (full state information concatenated over all timesteps) along with a high level description of the task (*e.g. the user wishes to water the plant*) and environment context (a description of the state of the environment) and query the LM for the likeliest feature $\phi_i$ that explains this difference[1]. Intuitively, $\tau_{R_{\theta^k}}$ acts as evidence for $\theta^k$ that is easier for the LM to

---
[1]If there are multiple missing features, the algorithm will identify them in future iterations.

**Algorithm 1:** ALGAE

> **Input:** Demonstrations $\mathcal{D} = \{\tau_i\}^{i=1...N}$
> **Init:** Features $\phi^0 = $ [distance to goal], confidence threshold $\epsilon$
> **while** *true* **do**
> > $\theta^k = \arg \max_\theta p(\mathcal{D} \mid \theta, \phi^k)$ // Optimize reward as in Eq. (3).
> > $\tau_{\mathcal{R}_{\theta^k}} = \arg \max_\tau \mathcal{R}_{\theta^k}(\tau)$ // Optimize trajectory according to this reward as in Eq. (1).
> > $\tau_H \sim \mathcal{D}$ // Sample one human demonstration to contrast it against.
> > **if** $\|\phi^*(\tau_{\mathcal{R}_{\theta^k}}) - \phi^*(\tau_H)\| < \epsilon$ *// If optimized trajectory explains the demo* **then**
> > > **break**
> > $\phi_i = \text{LM}(\tau_{\mathcal{R}_{\theta^k}}, \tau_H)$ // Query LM for missing feature that explains the contrast.
> > $\phi^{k+1} \leftarrow \phi^k \cup \phi_i$

interpret than reward weights, resulting in a sample from the LM's prior on $p(\phi_i \mid \tau_{R_{\theta^k}}, \tau)$. Thus, our approximate solution $\phi^{k+1}$ concatenates the LM sampled $\phi_i$ with the previous vector $\phi^k$.

**Querying for feature code.** Now that we have identified a missing feature $\phi_i$, we query the LM to directly generate the feature definition code that can be used to compute a feature value from raw state data over the given demonstrations. The LM is given raw trajectories $\tau$ and a natural language feature description of $\phi_i$, generated from the previous stage, and is asked to generate its code $\phi_i$. We append $\phi_i$ and its respective function code to the existing list of features $\phi^{k+1} \leftarrow \phi^k \cup \phi_i$.

### 4.2 Learning good reward functions

In the second stage, *reward validation*, we train and validate a reward function using the updated set of features $\phi^{k+1}$ as the basis function for the new reward, learning new parameters $\theta^{k+1}$. Because there may be additional missing features we have yet to identify, we now leverage existing user demonstrations to validate whether reward learning is complete. This iterative update is motivated by the desire to not *over-parameterize* the reward by specifying non-salient features.

**Learning a new reward.** Given our updated feature set $\phi^{k+1}$, we fix $\phi = \phi^{k+1}$ in Eq. (4) to obtain:

$$\arg \max_\theta p(\phi^{k+1}, \theta \mid \tau) = \arg \max_\theta \frac{p(\tau \mid \phi^{k+1}, \theta) p(\theta \mid \phi^{k+1}) p(\phi^{k+1})}{p(\tau)} = \arg \max_\theta p(\tau \mid \phi^{k+1}, \theta) \quad (6)$$

assuming uniform $p(\theta \mid \phi^{k+1})$. This equation directly corresponds to the MaxEnt IRL optimization procedure in Section 3 using $\phi^{k+1}$ as the feature vector. We now learn updated reward weights $\theta^{k+1}$ from the user demonstrations via Eq. (3), then optimize for an updated trajectory $\tau_{R_{\theta^{k+1}}}$ via Eq. (1).

**Validating the new reward.** To determine whether we have recovered all salient features, we leverage existing user demonstrations for validation. In a real-world deployment scenario, there are many reasonable ways to implement a comparison between trajectories, with the ideal being a human user giving feedback as to whether the robot is executing the desired behavior. However, in our simulated experiments, we approximate this judgement by measuring L2 distance between these trajectories in the ground truth feature space $\phi^*$. If this is within a threshold $\epsilon$, the learned reward function is sufficient for explaining the desired behavior and the algorithm terminates; otherwise, we iterate through the feature specification stage again. Pseudocode for the full pipeline can be seen in Algorithm 1.

## 5   Simulated Experiments

We evaluate ALGAE in both simulated and real-world robot environments. To highlight the flexibility of ALGAE across diverse feature types and high-dimensional control, our domains include 2D maze navigation tasks and a 7DoF JACO tabletop manipulation task, along with real-world mobile manipulation tasks with a Spot in Section 6. We use GPT-4o [24] to identify salient missing features in all experiments.

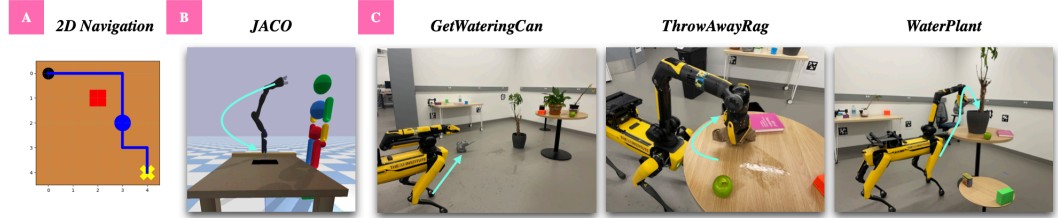

Figure 2: We evaluate on both simulated and real-world domains with a variety of missing features. **A**: 2D maze navigation, where the robot must navigate to a goal while interacting with other objects. **B**: 7DoF JACO manipulation, where the JACO arm must manipulate a held coffee mug while respecting features like *end effector orientation*. **C**: Spot mobile manipulation tasks, where the robot must complete tasks like *WaterPlant* while respecting the *height of pot*.

## 5.1 Experiment Setup

**Domain 1: 2D Navigation.** Inspired by the AI Safety Gridworlds [25], where under-specified features can implicitly lead to reward hacking behavior [26], we evaluate ALGAE on the following 2D maze navigation environments: GridRobot, Lavaland, and Island (Fig. 2). Trajectories are sequences of 9 states beginning with the start and ending at the goal. The 18-dimensional input consists of the $xy$ coordinate positions of the robot in the grid at each timestep. The action space is the four coordinate directions. Each scenario is a 5-by-5 2D grid consisting of an agent, a goal the agent must navigate to, and possible objects the agent must interact with. In GridRobot, the agent must avoid the (1) *Stove*; in Lavaland, the agent must avoid (2) *Lava*; in Island, the agent needs to drink (move to) (3) *Water*. To ensure there is the possibility of over-specifying features, we also randomly place non-relevant objects that do not impact the true reward in the environment (e.g., *Ball*).

**Domain 2: 7DoF Tabletop Manipulation.** To assess a higher-dimensional state and action space, we also evaluate a 7DoF JACO robot arm tasked with manipulating a coffee cup on a tabletop in a PyBullet simulator (see Fig. 2.) Each scenario is initialized with a starting and goal pose location, a laptop placed on the tabletop, and a human by the table. The robot must manipulate the coffee cup it is carrying to a specific goal location while respecting user preferences for motion. Trajectories are length 21, and each state has 97 dimensions: the $xyz$ positions of all robot joints and angles, and $xyz$ object positions along with rotation matrices. We introduce four scenarios based on the relevant human preference: (1) *Laptop*: $xy$-plane distance of the EE to the laptop, to ensure the cup does not pass over the laptop, (2) *Table*: $z$-distance of the EE to the table, to keep the mug of height close to the table, (3) *Orientation*: EE orientation relative to upright, to ensure the cup does not spill, and (4) *Human*: $xy$-distance of the EE to the human, to ensure the cup does not collide with the human.

**Test Scenarios.** To study whether our learned reward functions generalize across different scenarios related to spurious environment features, we construct two possible distribution shifts for each demonstrated scenario: (1) different trajectory goals (**Goal**) (e.g., the robot must move to a different goal location) and (2) different feature object locations (**Object**) (e.g., the laptop changes location in the scene). These are intended to test whether the learned reward captures true salient environment features rather than spurious feature correlations. We report the normalized reward of the optimized trajectory (against the reward of the ground truth best and worst trajectories) on a set of five randomly sampled test scenarios across three seeds.

**Comparisons.** A straightforward way to learn rewards from demonstrations is to learn directly from the state space. We refer to this baseline as (1) **Full-State**. We would expect such a baseline to fail to generalize to novel environments since the learned reward may be prone to overfitting spurious state correlations in the demonstrations, rather than true salient features in the environment. However, if access to a language prior was given without contrastive demonstrations (such that there is no demonstrated trajectory of the under-specified feature), we could query for all hypothesized features at once as (2) **LM-Feature**. This baseline is intended to isolate the impact of the language model proposing features given only the task description and environment context, and can be seen as

| | | GridRobot | Lavaland | Island | JACO | | | |
|---|---|---|---|---|---|---|---|---|
| | | *Stove* | *Lava* | *Water* | *Laptop* | *Table* | *Orientation* | *Human* |
| **Goal** | ALGAE | **1.00 (0.00)** | **1.00 (0.00)** | **0.99 (0.01)** | **0.90 (0.06)** | **0.68 (0.12)** | **0.73 (0.08)** | **0.66 (0.12)** |
| | Full-State | 0.49 (0.06) | 0.60 (0.06) | 0.52 (0.06) | 0.40 (0.05) | 0.37 (0.07) | 0.35 (0.03) | 0.39 (0.01) |
| | LM-Feat | 0.85 (0.02) | 0.86 (0.05) | 0.86 (0.08) | 0.53 (0.05) | 0.51 (0.16) | 0.58 (0.10) | 0.54 (0.04) |
| | LM-Rew | 0.81 (0.04) | 0.83 (0.10) | 0.72 (0.09) | 0.63 (0.08) | 0.48 (0.17) | 0.58 (0.07) | **0.67 (0.01)** |
| | Random | 0.58 (0.07) | 0.62 (0.04) | 0.57 (0.03) | 0.37 (0.03) | 0.39 (0.08) | 0.36 (0.03) | 0.38 (0.08) |
| **Obj.** | ALGAE | **1.00 (0.00)** | **1.00 (0.00)** | **1.00 (0.00)** | **0.93 (0.06)** | **0.65 (0.12)** | **0.67 (0.03)** | **0.53 (0.03)** |
| | Full-State | 0.62 (0.01) | 0.55 (0.06) | 0.44 (0.05) | 0.38 (0.07) | 0.31 (0.03) | 0.31 (0.06) | 0.44 (0.07) |
| | LM-Feat | 0.77 (0.04) | 0.81 (0.05) | 0.90 (0.05) | 0.54 (0.08) | 0.45 (0.14) | 0.59 (0.07) | 0.50 (0.01) |
| | LM-Rew | 0.76 (0.08) | 0.81 (0.10) | 0.79 (0.07) | 0.61 (0.06) | **0.56 (0.15)** | 0.58 (0.09) | **0.53 (0.02)** |
| | Random | 0.54 (0.02) | 0.61 (0.06) | 0.53 (0.02) | 0.38 (0.02) | 0.34 (0.09) | 0.36 (0.02) | 0.38 (0.08) |

Table 1: Normalized ground truth reward of the optimized trajectories produced by different methods across simulated test environments and feature scenarios. **Goal** represents test scenarios where the goal locations have changed from training and **Object** represents test scenarios where the relevant feature has changed location from training. We report standard error across three seeds.

a prompting-only baseline. For an additional prompting-only baseline, we also evaluate (3) **LM-Reward**, a baseline where the LM also proposes the reward weights in addition to the features. Inspired by work that proposes to construct rewards directly from language [27, 28, 29], this baseline is intended to test the zero-shot reward specification capability of LMs. Finally, to benchmark these values, we report a (4) **Random** reward, which randomly samples a trajectory for each test scenario.

## 5.2 Results and Analysis

**Single feature recovery.** We first assess the ability of ALGAE to recover a single missing feature. Table 1 shows results across test environments and their feature scenarios. As shown, ALGAE outperforms comparisons across all features. In simpler 2D environments such as GridRobot and Lavaland, ALGAE achieves close to perfect feature recovery and generalization on unseen test scenarios. While the two prompting-only baselines (LM-Feature and LM-Reward) perform well above Random, upon inspection of the LM rewards, they tend to produce many irrelevant features which result in *over-parameterized* reward functions. In the more complex JACO scenarios, ALGAE consistently outperforms baselines. Its performance is close to the prompting-only baselines on the *Orientation* task, which is unsurprising as there is no explicit penalty for over-parameterizing the reward (i.e., the robot can keep the cup upright while optimizing for additional features).

**Iterative feature recovery.** We next study the case where more than one feature is under-specified in a scenario, and ALGAE must recover them iteratively. In these experiments, the training scenario is instantiated with two missing features (*Water* and *Dinosaur* for **Island**) and (*Laptop* and *Orientation* for **JACO**). As seen in Fig. 3, ALGAE iteratively improves reward given multiple missing features.

## 6 Real-World Experiments

To assess the feasibility of our approach on robotic hardware operating in cluttered human environments, we now evaluate ALGAE's performance on mobile manipulation tasks with a Spot robot.

**Domain 3: Spot Mobile Manipulation.** Spot is a legged robot equipped with a manipulator (arm with gripper) and six RGB-D cameras (one in gripper, two in front, one on each side, one in back). Each scenario is initialized with a starting and goal robot pose location, along with the following objects: *watering can*, *plant pot*, *apple*, *towel*, and *textbook*. Trajectories are length 5 and each state consists of 10 dimensions: the $xyz$ positions of the robot and its hand, as well as rotation of the gripper. We introduce the following three scenarios and their relevant reward preferences: (1) *GetWateringCan* ($xy$-distance of the robot to the watering can), to ensure the robot is positioned to execute a successful grip, (2) *WaterPlant* ($z$-distance of the robot hand to the demonstrated watering

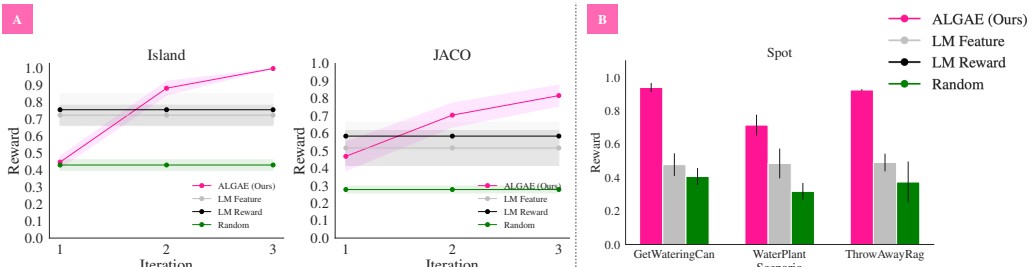

Figure 3: **A**: Normalized reward across multiple iterations in simulated domains. ALGAE (pink) improves across each iteration, continuously finding under-specified features and updating its reward estimate. In contrast, prompting-only baselines such as LM-Feature (gray) and LM-Reward (black) do not iteratively improve themselves after instantiation. **B**: Normalized reward on Spot domains. Real-world clutter introduced in the scene leads to over-parameterized LM-only rewards, increasing the gap between ALGAE and baselines. Error bars depict standard error across three seeds.

height above pot), to ensure the hand is adjusting to the pot size, and (3) *ThrowAwayRag* ($xy$-distance of the hand to the textbook), to ensure the robot does not drip water from the carried rag.

**Data collection.** For each scenario, a human teleoperates the robot from the starting to goal pose while satisfying the hidden preference (e.g. raise the robot hand to a particular height above the pot before performing a watering motion), and generates three demonstrations per scenario. We also collect all detected object identities and positions via image segmentation from the robot cameras [30, 31] and captioning [32] as environment context for the demonstrations.

**Test scenarios and comparisons.** We evaluate on the following test scenarios: for *GetWateringCan*, we vary the location of the watering can before executing a grip, for *WaterPlant*, we vary the height of the plant pot before executing a pour, and for *ThrowAwayRag*, we vary the location of the textbook on the table the robot hand must avoid while carrying the wet rag. To ensure the robot does not execute a drastically wrong action given an incorrectly learned reward, we constrain the sample space of the robot pose at test time around safe zones. We report the normalized reward of the optimized trajectory on a set of three test scenarios across three seeds. We report comparisons with the previous best performing method (**LM-Reward**) along with a **Random** benchmark.

**Results and Analysis.** ALGAE outperforms baselines on real-world scenarios (Fig. 3). This gap actually increases relative to simulated environments due to the real-world clutter introduced in the scene. Upon visual inspection of the LM-Reward features, we indeed see unnecessary features (e.g. relative position of the robot to *all* objects in the scene, not just the salient ones) included. This underscores the usefulness of ALGAE's interactive, iterative approach to feature construction.

## 7 Discussion and Limitations

ALGAE is an iterative reward learning framework that alternates between specifying missing features with a LM and weighting them into a reward via IRL. One limitation is that we used the ground truth feature vector to judge task completion on the contrasting pair of trajectories; future work might evaluate a true human-in-the-loop scenario, and study whether LM-informed priors have additional benefits beyond the performance improvements evaluated here. Moreover, we focus on reward functions that are linear in the feature set; in the future, we can imagine extending our method for nonlinear rewards. Finally, ALGAE is only effective for cases where the LM prior agrees with the desired behavior – which may not always be true in practice.

## Acknowledgements

We thank the MIT Learning and Intelligent Systems Group for providing robotic hardware resources used in this project. We thank members of the Language and Intelligence and Interactive Robotics Groups for helpful feedback and discussions. Andi Peng is supported by the NSF Graduate Research Fellowship and Open Philanthropy. Belinda Li is supported by National Defense Science and Engineering Graduate Fellowships. Ilia Sucholutsky is supported by a NSERC Fellowship (567554-2022). This work was partially supported by the National Science Foundation under grant IIS-2212310.

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

# A  Full Prompts

We provide the prompts used for both the feature query in all domains.

**2D Navigation.**

An agent is in a 2D gridworld environment of size 5x5. (0,0) is the top left corner and (4,4) is the bottom right corner (note this is not a typical coordinate grid).

The environment config has the following attributes: X (int): width of the environment Y (int): height of the environment starts (List[Tuple[int]]): starting coordinate of the agent goals_pos (List[Tuple[int]]): coordinate position of the goal goals_color (str): color of the goal goals_type (str): object type of the goal objects{1-3}_pos (List[Tuple[int]]): position(s) of other objects in the environment. This will be empty if there is no object in that slot. objects{1-3}_color (str): color of the corresponding object at the position objects{1-3}. This will be empty if there is no object in that slot. objects{1-3}_type (str): object type of the corresponding object at the position objects{1-3}. This will be empty if there is no object in that slot.

This is an environment config (disregard the extra attributes that are not specified above): {env}

The following is a demonstration the agent took in this first environment (each trajectory contains x,y coordinates for the agent, 8 timesteps total). Keep track of where the agent's positions relative to the other objects in the environment. For example, [0,0 0,1 0,2 0,3] means the agent started at [0,0] and moved to [0,3]. If there is an object at [0,3], the agent moved towards that object.

demonstration = {demo}

The following is an (incorrect) demonstration that the agent took in this environment. The trajectory is the same format as the first demonstration. incorrect trajectory = {best_traj}

Describe the demonstration and trajectory in the context of the environment they are in, then use this to give the specific feature in that environment that explains what the demonstration did better. Reason step by step and think about why the agent might be moving in the way that it is. If there are multiple of the same object, please give the average distance to all objects of that type. Please specify a feature (only one feature e.g. if there are multiple possible explanations choose the most likely object) that we can write code to compute from the position of the agent to other objects in the scene.

This is the current list of features that we already know: {self.feature_names}. Please give a different feature in your answer (e.g. if distance to goal feature is already in the list, do not give distance to goal again). Please give your confidence for your answer between 0 and 1. Ground your answer in visible objects in the scene, and give an answer (only 1 specific answer (e.g. distance to book objects)) that we can compute relative to the position of the agent.

**JACO.**

A 7-DoF Jaco robot arm is carrying a coffee cup on a tabletop environment. The arm is carrying a fragile ceramic coffee cup that is at risk of slipping. The arm is controlled via a Pybullet simulator.

The environment config has the following attributes: object_centers: {'HUMAN_CENTER': [x,y,z], 'LAPTOP_CENTER': [x,y,z]} # coordinates of the human

and laptop objects in the environment tabletop: z=0 # the z-coordinate plane of the tabletop

This is an environment config (disregard the extra attributes that are not specified above): {env}

The following is a correct demonstration trajectory the robot took in this first environment. Each trajectory is a 21x97 matrix representing the 21 timesteps the robot took over the course of the demonstration. Each state in the timestep is a 97-dimensional vector representing the x,y,z positions of all robot joints and objects, and their rotation matrices. This is the code that generates each state. Note the last six values represent fixed xyz positions of the human and laptop:

```
    # Get relevant objects in the environment
    posH, _ = p.getBasePositionAndOrientation(objectID["human"])
    posL, _ = p.getBasePositionAndOrientation(objectID["laptop"])
    object_coords = np.array([posH, posL])

  # Get xyz coords and orientations. 'waypt' is a single waypoint in the trajectory
    move_robot(objectID["robot"], joint_poses=waypt)
    coords = robot_coords(objectID["robot"])
    orientations = robot_orientations(objectID["robot"])
    return np.reshape(np.concatenate((waypt[:7], orientations.flatten(),
    coords.flatten(), object_coords.flatten())), (-1,))
```

Reason step by step, and keep track of where the robot's movements relative to important features in the scene. correct trajectory = {demo}

The following is an incorrect trajectory that the robot took in this environment. The trajectory is the same format as the first demonstration. incorrect trajectory = {best_traj}

Describe this trajectory in the context of the environment it is in, then use this to give the specific feature in that environment that explains why this trajectory was incorrect relative to the demonstration, which was correct. Reason step by step and think about how the robot's movements are related to the objects in the environment, as well as how aspects of the robot's position and orientations may be impacting the desired task (bringing coffee). Please specify a feature (only one feature e.g. if there are multiple possible explanations choose the most likely object) that we can write code to compute from the position of the robot to other objects in the scene. This is the current list of features that we already know: {self.feature_names}. Please give a different feature in your answer (e.g. if distance to goal feature is already in the list, do not give distance to goal again). Please give your confidence for your answer between 0 and 1. Ground your answer in visible objects in the scene, and give an answer (only 1 specific answer (e.g. distance to book)) that we can compute from the position of the robot.

**Spot.**

This is an environment configuration of objects and their xyz positions in the room.

Environment: {environment}

Goal: {goal}

The following is a successful trajectory (over 5 timesteps) the robot took in this environment (each trajectory contains xyz positions of the robot, along with

the xyz positions and rotation quarternion of the robot's hand).
Keep track of where the robot's positions and hand are relative to the other objects in the environment.

Successful trajectory: {demo}

The following is an unsuccessful trajectory (over 5 timesteps) the robot took in this environment.

Failed trajectory: {failed traj}

Describe the key feature that distinguishes the successful trajectory from the failed trajectory. Make sure the feature is semantically sensible for the goal. Reason step by step. Please specify a feature (only one feature e.g. if there are multiple possible explanations choose the most likely one). Make sure the feature is computable from the state features in the scene (and no other information): plant_XYZ, robot_XYZ, robot_arm_XYZ, robot_arm_rotation. The feature can be relative positions to objects, relative positions to positions around objects (e.g. a force field around the object), relative positions to absolute points in space, relative rotations, etc. The feature can even be non-linear. However, make sure the feature is continuous and not discrete. Please describe the feature in detail.

