# OpenReview forum: "Adaptive Language-Guided Abstraction from Contrastive Explanations"
_robot-learning.org/CoRL/2024/Conference — CoRL 2024_

### Official Review · Reviewer_cr8T · 2024-07-12
**Very compelling and generally useful inverse RL algorithm using language model**

**Originality:** 4
**Technical Quality:** 4
**Clarity Of Presentation:** 4
**Potential Impact:** 3
**Recommendation:** 3
**Confidence:** 4

**Review:**

The authors present a very compelling story for reward selection. It's a welcome addition to add more to the reward learning with LM literature.

The most compelling pieces is the iterative feature addition and the evaluation/weighting of the generated reward function. In prior reward learning work [1], weighting was often ignored and reward function generation was basically considered a zero-shot process. There was no iteration once the reward function was generated, so this iterative process lets you craft the best reward given a set of demonstrations. I think this system is very valuable, and likely to be helpful in constructing reward functions from language models.

The experiments are promising, but it would be nice to see some more results here. The core hypothesis is that human-crafted feature selection is hard, and it's prone to reward hacking, so the priors in the language model can propose the most salient features for the task. However, the state representation for all the experiments were highly handcrafted already (the positions of the robot and objects). It is unlikely that the policy will focus on spurious features that are not relevant once there is already a small state space (<50 values). By handcrafting state features, you are already doing some steering of what kind of reward function the language model should generate.

For the purpose of this paper which is proving the ALGAE method, this is alright. But it would be nice to show an example where the state representation might be a full image, and higher level features could be extracted from the VLM (via tool use) when computing the reward function [2,3].

The gridworld experiments are a good unit test, and it's good to prove that ALGAE outperformed baseline methods in that toy scenario. The sim robot experiment and real robot experiment were a bit nicer, but like I mentioned above, it would be nicer if the state representation wasn't already highly reduced.

The method is very general, and the contrastive method of evaluating the reward function is very appealing. You could even have some holdout demonstrations from your dataset if you didn't want to take a demonstration in-the-loop.

[1] Language to Rewards for Robotic Skill Synthesis, Yu et al
[2] ViperGPT: Visual Inference via Python Execution for Reasoning, Suris, Memon, Vondrick
[3] How to Prompt Your Robot: A PromptBook for Manipulation Skills with Code as Policies, M.G. Arenas et al

**Quality Of The Limitations Section:**

3

**Questions For Rebuttal:**

- The reward function building adds new features at every iteration. If a bad feature is proposed, how could we determine if we should not include it? Or is that implicit in the feature weighting.
- A small detail: was the feature weighting a simple linear combination of weights, or something more complex in your experiments?
- The biggest limitation is most likely the simple state representation. Could you comment on how to remove some of the handcrafting of the state representation when doing reward function creation?

**Robotics Focus:**

4

**Summary Of Paper:**

The main contribution is a method for inverse RL that uses a language model to determine reward features that are relevant for the task.

**Summary Of Recommendation:**

Overall, the work is quite nice. The iterative construction and evaluation of the reward function is a good contribution. Would love to see some more real robot experiments with a bit less state tuning too.

---

### Official Review · Reviewer_vt5Y · 2024-07-19
**novel idea for feature learning in IRL**

**Originality:** 3
**Technical Quality:** 3
**Clarity Of Presentation:** 4
**Potential Impact:** 3
**Recommendation:** 3
**Confidence:** 4

**Review:**

Strengths:
- The idea of leveraging pretrained language models for iterative feature learning in IRL is novel
- The proposed method cleanly decouples the reward learning and feature learning problem
- Incorporating large language model functionality into traditional IRL method combines the strengths of data-driven and model-based methods
- Experiments in this work contain different domains in both sim and real
- The proposed method is compared with ablations
- The writing and presentation of this paper is clear and easy to follow

The major weakness of this work is that the experiments are heavily engineered:
- In all tasks, there are at most 3 features to learn, which favors the proposed method that learns one feature at a time
- One baseline could be randomly sampling a subset of features and pick the one that best explain the demos
- Current experiments assume access to known GT feature vectors for determining if the current set of features are good enough, which the authors acknowledge as a major limitation but do not practically address or demonstrate the performance of the system without such assumption
- The prompt design seems to involve a lot of domain knowledge and tuning effort
- if these tasks are easy enough to hand engineer rewards, why use LLMs to learn?

**Quality Of The Limitations Section:**

1

**Questions For Rebuttal:**

- [see comments on weakness above]
- Can you show an example input trajectory to the LLM and its response? How do you make sure if LLM can understand the trajectory in the form of numbers/matrices? Does the output of the LLM end up writing code to parse the trajectory information or it only uses text?
- On a similar note to the question above, why not use visual state as input for LLM? For the LM-feature ablation, I can imagine if the LLM does not see the layout of the environment, it may want to track each and every object for obstacle avoidance; but if it could see the image and understand what is near the robot and target object, it might filter unrelated features
- How does the proposed method compare with zero-shot using language models to plan robot behavior? If it is able to understand trajectories in numbers, seems like it can also be prompted to output the trajectory directly. Similarity, is VoxPoser [1] a fair baseline here?

[1] https://voxposer.github.io/

**Robotics Focus:**

4

**Summary Of Paper:**

This paper proposes to leverage language models to learn missing features for inverse reinforcement learning. The reward learning problem is decoupled into two sub problems: 1) feature learning with language model reasoning about one missing feature at a time; and 2) reward learning using demonstrations and learned features. Experiments in both sim and real domains show the effectiveness of the proposed method in comparison with baseline and ablated methods.

**Summary Of Recommendation:**

The method is novel and interesting but the experiments are toy and more baselines should be included.

---

### Official Review · Reviewer_sFkR · 2024-07-21
**Submission422 Review**

**Originality:** 3
**Technical Quality:** 3
**Clarity Of Presentation:** 3
**Potential Impact:** 3
**Recommendation:** 3
**Confidence:** 3

**Review:**

The authors introduce a method called ALGAE, which aims to improve robot learning by leveraging human-like priors encoded in natural language. The method addresses the challenge of learning robust reward functions from a limited set of human demonstrations by identifying and incorporating relevant features into the reward learning process.
The authors provide a detailed explanation of the method and support their claims with empirical evidence from experiments conducted in both simulated and real-world environments.
The paper is well written generally and and the use of figures and diagrams effectively show the contribution.  However, certain technical sections, especially those involving equations, could benefit from more intuitive explanations.
The paper does not thoroughly discuss limitations or failure cases of the method. Addressing these aspects could provide a more deatailed view of the method's applicability and robustness in various scenarios. more detailed comparisons with a wider range of existing method, would help to better highlight the unique advantages of the approach

**Quality Of The Limitations Section:**

2

**Questions For Rebuttal:**

How well does ALGAE generalize to entirely new tasks (or environments) that were not seen during the training phase?

**Robotics Focus:**

4

**Summary Of Paper:**

The authors introduce a method called ALGAE, which aims to improve robot learning by leveraging human-like priors encoded in natural language. The method addresses the challenge of learning robust reward functions from a limited set of human demonstrations by identifying and incorporating relevant features into the reward learning process.

**Summary Of Recommendation:**

I recommend an accept

---

### Decision · Program_Chairs · 2024-09-04

**Decision:**

Accept

**Comment:**

The paper's strengths include:
The ALGAE method, which leverages language models for feature learning in inverse reinforcement learning (IRL), offers a novel approach by iteratively learning and integrating features relevant to the reward function.
The separation of feature learning and reward learning into distinct subproblems helps in focusing on each component effectively, combining the strengths of data-driven and model-based approaches.
The paper provides empirical evidence of the method's effectiveness across both simulated and real-world environments, demonstrating clear advantages over baseline methods and ablated versions.
The writing is clear and well-organized, making the method and its results easy to understand. The iterative feature addition and evaluation process is a valuable contribution to the field.

The paper's weaknesses include:
The experiments are somewhat engineered with a focus on a small number of features, which may not fully showcase the method's robustness in more complex scenarios. More diverse baselines and experiments are needed.
The experiments utilize highly handcrafted state representations, which could bias the results. More exploration with less engineered state representations or full image inputs would be beneficial.
The paper does not thoroughly address limitations or failure cases, such as generalization to new tasks or environments and reliance on known ground truth feature vectors.
The design of prompts for language models involves significant domain knowledge and tuning, which might not be practical for all applications.

Post-rebuttal meta-review:
The paper introduces ALGAE, a method designed to improve robot learning by leveraging language models to identify and incorporate relevant features into the reward learning process, particularly in scenarios with limited human demonstrations.

ALGAE uses language models to iteratively identify human-meaningful features, which are then used to guide the reward learning process. The method alternates between feature identification and reward optimization, allowing it to refine its understanding and adapt to missing features without human intervention. ALGAE demonstrates its ability to generalize learned reward functions across various simulated and real-world robotic environments.

The integration of language models into feature learning for inverse reinforcement learning (IRL) is a novel approach. The paper is well-written, clearly explaining the method and supporting its claims with solid empirical evidence in both simulation and real-world experiments. The method shows robustness in extracting meaningful features and generalizing across different tasks.

The experiments appear heavily engineered, with a focus on a small number of features, which may not fully showcase the method's robustness in more complex scenarios. The reliance on highly handcrafted state representations could bias the results, limiting the exploration of more diverse or less engineered inputs. The paper lacks a deep comparative analysis with state-of-the-art methods, which could better demonstrate its distinctiveness and impact.

Final Recommendation: Borderline Accept-- The paper presents a novel approach with promising results, but it has some limitations, such as engineered experiments and a lack of comprehensive baselines, which may affect its broader applicability. While it is a valuable contribution, further work could strengthen its claims.